# Quantifying the impact of early life growth adversity on later life health

## Abstract

**Background** Early-life growth adversity is important to later-life health, but precision assessment in adulthood is challenging. We evaluated whether the difference between attained and genotype-predicted adult height ("height-GaP") would associate with prospectively ascertained early-life growth adversity and later-life all-cause and cardiovascular mortality.

**Methods** Data were first analyzed from the Avon Longitudinal Study of Parents and Children (ALSPAC; $n = 4582$; 56/43% female/male) and UKBiobank ($n = 483,385$; 54/46% female/male). Genotype-predicted height was calculated using a multi-ancestry polygenic height score. Height-GaP was calculated as the difference between measured and genotype-predicted adult height. Early-life growth conditions were ascertained prospectively via standardized procedures (ALSPAC) and mortality via death register (UKBiobank). Regression models examined: (i) adult height-GaP as the outcome with early-life growth conditions as predictors; and (ii) mortality as the outcome with adult height-GaP as predictor. All models were adjusted for age, sex, genotype-predicted height and genetic ancestry. Analyses were replicated in the Dunedin Multidisciplinary Health and Development Study (DMHDS; $n = 855$; 49/51% female/male) and the Multi-Ethnic Study of Atherosclerosis (MESA; $n = 6352$; 52/48% female/male).

**Results** Here we show that among ALSPAC participants (median [IQR] age: 24 [18-25] years at height-GaP assessment), lower gestational age at birth, greater pre- and post-natal deprivation indices, tobacco smoke exposure and less breastfeeding are associated with larger adult height-GaP deficit ($p < 0.01$). Among UKBiobank participants (mean ± SD age: 56 ± 8 years at height-GaP assessment), height-GaP deficit is associated with death from all-causes (adjusted hazard ratio comparing highest-to-lowest height-GaP deficit quartile [aHR]: 1.25 95%CI: 1.21–1.29), atherosclerotic cardiovascular disease (aHR: 1.32 95%CI: 1.23–1.42) and coronary heart disease (aHR: 1.64 95%CI: 1.49-1.81). Early- and later-life height-GaP associations replicate in DMHDS and MESA.

**Conclusions** This study introduces a precision index of early-life growth adversity deployable in adulthood to investigate the developmental origins of longevity and improve health equity across the life course.

## Plain language summary

Early-life exposures that adversely affect body growth have been implicated in later-life health, but assessment in adulthood is challenging. We tested whether the difference between one's standing height in adulthood and one's genetically predicted adult height ("height-GaP") would be a suitable method for assessing cumulative exposure to early-life growth adversity and predicting later-life health. In two early-life studies that followed children into adulthood, pre-natal and post-natal exposure to tobacco smoke, poverty, less breast feeding and poor nutrition were associated with larger adult height-GaP deficit. In two adult studies, larger height-GaP deficit was associated with death from all-causes and from heart disease. This study introduces a precision measure of early-life growth adversity that can be deployed in adulthood to investigate the developmental origins of health and disease.

The period of human ontogenetic growth, which spans conception to late adolescence, is increasingly recognized as an important window of susceptibility to exposures and events that influence later-life health[1]. Transforming this understanding into substantive reductions in disease burden and health inequity is challenging, in part, due to the lack of simple methods to quantify early-life growth adversity in adulthood. These challenges include the diverse and often correlated nature of early-life factors affecting growth, recall and selection bias when ascertained retrospectively, and the long latency to later-life health outcomes. This study sought to evaluate a precision index of early-life growth adversity deployable in adulthood to facilitate life course research on the developmental origins of health and disease.

✉ e-mail: benjamin.m.smith@mcgill.ca; bs2723@cumc.columbia.edu

Anthropometric indices of early-life growth adversity readily measured in adulthood–such as measured height, sitting-to-standing-height ratio and leg-to-torso length ratio[2–5]–are determined in part by genetics[6–8], reducing their utility as a quantitative marker of non-genetic early-life growth adversity. Molecular indices of early-life growth adversity such as adult telomere length and DNA methylation signatures (e.g., epigenetic age) are limited by their continued plasticity in adulthood[9–15].

Human height increases during the period of ontogenetic growth and is determined in part by genetics and by early-life growth conditions. A recent genome-wide association study of ~5.4 million adults reported a saturated map of common genetic variants associated with adult height that accounts for over 90% of trait heritability and up to 45% of trait variance[6]. It follows that the difference between measured and genotype-predicted adult height, here referred to as "height-GaP", may represent a simple index of early-life growth adversity. If true, such an index could (i) facilitate discovery of threats to human development, (ii) serve as a surrogate endpoint for early-life interventions aiming to improve later-life health, and (iii) improve adult risk stratification and endo-phenotyping for diseases with developmental origins.

This study first demonstrates that prospectively ascertained early-life factors known to influence ontogenetic growth associate with adult height-GaP. The study then demonstrates that adult height-GaP is associated with later-life mortality from all-causes and from atherosclerotic cardiovascular disease, the latter being a major public health burden with established developmental origins[1,16–33].

## Methods
### Design
Cross-sectional and longitudinal analyses of cohort data.

### Data
The Avon Longitudinal Study of Parents and Children (ALSPAC) is a prospective birth cohort that enrolled women who were pregnant in Avon, England with expected delivery dates between April 1 1991 and December 31 1992[34–36]. Initial recruitment of 14,541 pregnancies resulted in 13,988 children alive after 1 year. Following identification of additional eligible children born between April 1 1991 and December 31 1992 in Avon, England, further recruitment attempts at follow-up years 7 and 8 resulted in a total of 15,447 pregnancies included in the study, with 14,901 children alive after 1 year. Parents and children have been characterized over the past four decades including anthropometry, questionnaires and genotyping. ALSPAC provides a searchable data dictionary online[37]. For this analysis, we included participants with measured standing height performed at visit year 18 or 24, genotype data and information on at least one prospectively ascertained early-life growth condition ($n = 4582$). Ethical approval for the study was obtained from the ALSPAC Ethics and Law Committee and the Local Research Ethics Committees.

UKBiobank (UKB) is a prospective cohort that enrolled over 500,000 adults aged 40–69 years between 2006 to 2010 across England, Scotland and Wales[38]. Participants were extensively characterized at baseline including standardized anthropometry, questionnaires, genotyping, with follow-up assessment of vital status. For this analysis, we included participants with measured adult standing height, genotype data and information on vital status ($n = 483,385$).

The Dunedin Multidisciplinary Health and Development Study (DMHDS) is a population-based birth cohort of 1037 participants born between April 1972 and March 1973 at Queen Mary Maternity Hospital in Dunedin, New Zealand[39,40]. Participants were enrolled at 3 years old and have undergone periodic standardized assessments into the fifth decade of life. For this analysis, we included participants with measured adult standing height, genotype data, and at least one prospectively ascertained early-life growth condition ($n = 855$).

The Multi-Ethnic Study of Atherosclerosis (MESA) is a United States multi-site cohort that enrolled 6814 adults 45 to 84 years old between 2000 and 2002 who self-identified as White, African American, Hispanic or

Chinese. Exclusion criteria included clinically apparent cardiovascular disease and impediments to long-term follow-up. For this analysis, we included participants with measured standing height at baseline, genotype data, and information on vital status ($n = 6352$).

Institutional research ethics board approval was obtained for this study (McGill University Health Centre, 10644) and for each cohort (ALSPAC: approval for the study was obtained from the ALSPAC Ethics and Law Committee [a list of approval numbers can be found here: https://www.bristol.ac.uk/alspac/researchers/research-ethics/]; UKBiobank: the North West Centre for Research Ethics Committee (11/NW/0382) [https://www.ukbiobank.ac.uk/about-us/how-we-work/ethics/]; DMHDS: the New Zealand Health and Disability Ethics Committee; MESA: Wake Forest University, IRB00008492; Columbia University, IRB00002973; Johns Hopkins University, IRB00001656; University of Minnesota, IRB00000438; Northwestern University, IRB00005003; University of California Los Angeles, IRB00000172; University of Washington, IRB00005647); all participants provided written informed consent.

### Measured standing height
All cohorts measured standing shoeless height according to standardized protocols with calibrated stadiometers.

### Genotyping
ALSPAC genotyping was performed via Illumina 660 quad sequencing with imputation according to the 1000 Genomes reference panel[35]. UKBiobank genotyping was performed using the UKBiobank Axiom Array with imputation according to a combination of the Haplotype Reference Consortium and 1000 Genomes reference panels[38]. DMHDS genotyping was performed using the Illumina HumanOmni Express 12 BeadChip array with imputation using the 1000 Genomes phase 3 reference panel[41]. MESA genotyping was obtained through TOPMed Freeze 10b, with quality control performed based on existing TOPMed protocols[42].

### Genotype-predicted adult height
Genotype-predicted height was computed using the all-ancestry polygenic height weights reported by Yengo et al.[6]. Each participant's polygenic height score was first calculated as the weighted sum of height-increasing alleles (additive model; missing effect alleles were assigned the effect allele frequency of the participant's respective cohort). Genotype-predicted height was then computed by fitting cohort-, sex- and race/ethnic-specific linear regression models of measured height. Sensitivity analyses computed (i) polygenic height score using ancestry-specific weights[6], and (ii) computed genotype-predicted height using cohort- and sex-specific linear regression models of measured height.

### Height-GaP
Height-GaP was computed as the difference of measured adult height minus genotype-predicted height in centimeters. A lower height-GaP value represents a larger deficit in measured height compared to genotype-predicted height.

### Early-life growth conditions in ALSPAC and DMHDS
Birthweight was extracted from medical records in ALSPAC and DMHDS; birth length was measured by trained ALSPAC staff and extracted from medical records in DMHDS[43,44]. Gestational age was estimated from the date of last menstrual period and confirmed via ultrasound in ALSPAC, and extracted from medical records in DMHDS[43]. Standardized questionnaire items were used to assess breastfeeding status and duration (months), maternal smoking during pregnancy (cigarettes/day) and household tobacco smoke exposure in ALSPAC and DMHDS[34,45]. Dietary patterns in ALSPAC were assessed using standardized questionnaires and summarized using principal components analysis[46]. Residential address information during pregnancy, at birth, and at post-natal ages 1–12 years were used to compute neighborhood-level indices of multiple deprivation during in ALSPAC, with higher values corresponding to greater deprivation[47]. In

DMHDS, socioeconomic status was quantified with a six-point occupational measure that was most widely used in the New Zealand research community while the participants were growing up[48]. The variable used in our analyses, childhood socioeconomic status, is the average of the highest socioeconomic status level of either parent, assessed repeatedly at the participant's birth and at ages 3, 5, 7, 9, 11, 13, and 15 years.

## Mortality in UKBiobank and MESA

Information on vital status was ascertained in UKBiobank via linkage to the British National Death Registry and in MESA via 9–12 month interval participant residence telephone contacts and linkage to the National Death Index of the National Vital Statistics System yielding date of death and the International Classification of Disease-(ICD)−10 code listed as the primary (underlying) cause of death[49,50]. Time until death or censorship was computed as the difference in years between death or last study contact and baseline study visit.

Deaths due to atherosclerotic cardiovascular disease (fatal coronary heart disease or fatal stroke) and atherosclerotic coronary heart disease were defined in UKBiobank based on the ICD-10 code listed as the underlying (primary) cause of death (I20-25, I60-61, I63-64)[51] and in MESA by standardized adjudication that included paired cardiologist or neurologist review of abstracted medical records, with disagreements resolved by full committee review[52].

## Other variables (see Supplementary Data 1 for additional details)

Age, sex and race or ethnicity were self-reported. Principal components of genetic ancestry were derived from genotype data in UKBiobank, DMHDS and MESA[38]. The ALSPAC sample consisted only of persons of European ancestry defined by genetic principal components[53]. In UKBiobank and MESA, cigarette smoking status, pack-years of smoking, alcohol use status, average number of drinks per week, quantity of moderate-to-vigorous physical activity per week, household income, educational attainment and health insurance status (in MESA only) were derived from standardized questionnaire items. Diabetes status was defined by a fasting blood glucose ≥7.0 mmol/L (126 mg/dl) or diabetes medication use; low-density lipoprotein cholesterol concentration was derived from fasting blood sample triglyceride concentration. Hypertension status was defined by standardized automated measures of systolic and diastolic blood pressure or the use of anti-hypertensive medication.

For height loss sensitivity analyses, annualized longitudinal change in measured height was computed in UKBiobank as the difference in height in cm between baseline and follow-up assessment divided by the time interval between assessments in years.

## Height-loss sensitivity analysis

To evaluate later-life height loss as a potential confounder of the later-life height-GaP mortality associations, we first estimated the sex-specific age at which height loss was first evident using longitudinal height measurements in UKBiobank. Next, we estimated the age- and sex-specific percentile distribution of annualized rate of height loss from the age of onset using quantile regression. Finally, we corrected each participant's measured height (and thus height-GaP) back to the age of height loss onset using (i) the median annualized height loss and the participant's age and sex, and (ii) a "worst case" scenario of height-loss confounding. Under the "worst case" scenario, on a percentile-by-percentile basis, a participant with larger height-GaP deficit was assumed to have experienced greater later-life height loss. For example, participants in the 95th percentile of height-GaP deficit are assumed to have experienced the 95th percentile of annualized height loss, and their measured height (and thus height-GaP) is corrected accordingly. This procedure is repeated on a percentile-by-percentile basis. Height loss-adjusted height-GaP values were then used to compute mortality associations adjusting for the same covariables listed above.

## Statistics and reproducibility

Participant characteristics are summarized by quantile of adult height-GaP. The hypothesized causal relationships of early-life growth conditions with later-life health outcomes are depicted in a directed acyclic graph (Supplementary Fig. 1). Associations were first computed using ALSPAC and UKBiobank data, followed by replication analyses in DMHDS and MESA.

Separate generalized linear and spline regression models of adult height-GaP were fit for each prospectively ascertained early-life growth condition. Sample sizes were $n = 4582$ and $n = 855$ for the ALSPAC and DMHDS cohorts, respectively. Models were unadjusted and adjusted for age at height-GaP assessment, sex and genotype-predicted height. Early-life associations were not further adjusted for principal components of genetic ancestry due to the ALSPAC cohort's prior sampling according to principal components. Tabular results are reported per 1-SD or quantile contrast depending on whether the linear or spline model fit was superior, assessed according to model Akaike Information Criterion. Heterogeneity of associations by sex were evaluated by adding a height-GaP-sex product term to each regression model. The statistical significance threshold for ALSPAC analyses was a two-sided $p$-value = 0.010 to account for testing of five early-life growth categories (nutrition [breastfeeding, diet], peri-natal [gestational age at birth, birth weight, birth length], socio-economic deprivation [multiple deprivation index], and noxious exposures [tobacco smoke, $PM_{2.5}$]. The threshold for association analyses in the DMHDS replication sample, which was similarly not adjusted for principal components of genetic ancestry, was a two-sided $p$-value = 0.050. Imputation of missing data using all analysis variables as predictors in ALSPAC and DMHDS was performed using polytomous logistic regression and predictive mean matching for categorical and continuous variables, respectively, across 100 imputed datasets ('MICE' R package, v 3.16.0).

Proportional hazard models of mortality (all-cause and cause-specific) were fit with baseline height-GaP as the independent variable of interest. Sample sizes were $n = 483,385$ and $n = 6352$ for the UKBiobank and MESA cohorts, respectively. Model 1 adjusted for age, sex and principal components of genetic ancestry. Model 2 (main model) additionally adjusted for genotype-predicted height to provide the model with a linear combination of information equivalent to measured height. Heterogeneity of associations by sex and by race-ethnicity were evaluated by including main effect and product terms in regression models; the race-ethnicity interaction models excluded principal components of genetic ancestry. The statistical significance threshold in UKBiobank was a two-sided $p$-value = 0.017 (0.05/3) to account for testing three mortality outcomes; the threshold for association analyses in the MESA replication sample was a two-sided $p$-value = 0.050. Mortality associations with measured height and genotype-predicted height were also assessed using the aforementioned models.

**Sensitivity analyses**. To evaluate later-life height loss we corrected each participant's measured height (and thus height-GaP) back to the age of height loss onset using (i) the median annualized height loss and the participant's age and sex, and (ii) a "worst case" scenario of height-loss confounding (See *Methods: Height-loss sensitivity analysis* for details).

To account for potential confounding of the height-GaP association with later-life mortality by adult health-related factors, the main model was additionally adjusted for the following: baseline hypertension status, systolic blood pressure, diabetes status, weight status, cigarette smoking status (current, former, never), pack-years of smoking, alcohol consumption status and drink frequency, minutes of weekly moderate and weekly vigorous physical activity, educational attainment, household income and, in MESA, health insurance status. We note, however, that these adult health conditions have been implicated in the developmental origin of health and disease paradigm and may represent partial or full mediators[1].

Finally, ALSPAC analyses were repeated among participants with complete data and among participants with height measured at the 24 year study visit, and UKBiobank and MESA analyses were repeated using ancestry-specific polygenic height scores and using genotype-predicted

height computed via cohort- and sex-specific linear regression models of measured height.

All analyses were performed using R version 4.4.1 and Python version 3.12.5.

## Results

Characteristics of the 4582 ALSPAC participants included in this analysis are summarized in Supplementary Data 2. The median (IQR) age at the time of height-GaP assessment was 24 (18, 25) years, 56% were female and the mean ± SD measured height was 180 ± 7 cm for males and 166 ± 6 cm for females. The polygenic height score accounted for 40.3% and 37.6% of the variance in adult height among males and females, respectively, and the mean ± SD height-GaP was 0.0 ± 5.0 cm (95th percentile range: −9.9 to 9.7 cm). Characteristics of excluded participants are summarized in Supplementary Table 1.

Characteristics of the 483,385 UKBiobank participants included are summarized in Supplementary Data 3. The mean ± SD age at height-GaP assessment was 56 ± 8 years, 54% were female, mean ± SD measured height was 176 ± 7 cm for males and 163 ± 4 cm for females, 10.5% were current smokers and 34.5% were former smokers (median [IQR] 19 [10, 32] pack-years among ever smokers). The polygenic height score accounted for 37.1% and 35.2% of variance in measured height among males and females, respectively, and the mean ± SD height-GaP was 0.0 ± 5.2 cm (95th percentile range: −10.1 to 10.5 cm). Over a median of 12.4 years of follow-up (5,996,608 person-years), there were 35,703 deaths, of which 7177 were attributed to atherosclerotic cardiovascular disease and 3801 to atherosclerotic coronary heart disease. Participant characteristics were generally similar across height-GaP deficit quartile, except those with larger height-GaP deficit tended to be older, to have ever smoked and to have a higher number of pack-years. Excluded participant characteristics are summarized in Supplementary Data 4.

Characteristics of the replication cohorts are presented in Supplementary Table 2 *and* Supplementary Data 5. The DMHDS sample included 855 participants (age: 26 ± 1 years at height-GaP assessment; 49% female; height: 178 ± 6 cm for males and 165 ± 6 cm for females) with a height-GaP of 0.0 ± 5.3 cm (95th percentile range: −9.9 to 10.3 cm). The MESA sample included 6352 participants (age: 62 ± 10 years at height-GaP assessment; 52% female; measured height: 173 ± 8 cm for males and 160 ± 7 cm for females) with a height-GaP of 0.0 ± 5.7 cm (95th percentile range: −11.1 to 11.5 cm). Self-identified race-ethnic proportions were 39.1% White, 26.2% African American, 22.6% Hispanic, and 12.1% Chinese. Over a median of 15.0 years of follow-up (92,319 person-years), there were 1,337 deaths, of which 233 and 155 were adjudicated as being due to atherosclerotic cardiovascular and coronary heart disease, respectively.

### Early-life growth conditions and adult height-GaP

The adjusted associations of early-life growth conditions with adult height-GaP in ALSPAC are summarized in Fig. 1 and Supplementary Table 3.

In the main adjusted model, larger adult height-GaP deficits were observed among participants with greater levels of multiple deprivation during pregnancy (mean height-GaP difference per quintile of multiple deprivation: −0.16 cm; 95%CI: −0.28 to −0.05 cm; p = 0.006), greater maternal smoking during pregnancy (mean height-GaP difference comparing 0 vs. 20+ cigarettes per day: −1.51 cm; 95%CI: −2.71 to −0.30 cm; p = 0.001), lower maternal "healthy diet" principal component during pregnancy (mean height-GaP difference per 1-SD decrement in "healthy diet": −0.43 cm; 95%CI: −0.59 to −0.27 cm; p < 0.001), lower gestational age at birth (mean height-GaP difference comparing <32 weeks to 38+ week: −4.09 cm; 95%CI: −7.14 to −1.03 cm; p = 0.005), lower birth weight (mean height-GaP difference per 1-kg decrement: −2.22 cm; 95%CI: −2.63 to −1.82 cm; p < 0.001), lower birth length (mean height-GaP difference per 1-cm decrement: −0.67 cm; 95%CI: −0.78 to −0.55 cm; p < 0.001), less breastfeeding (mean height-GaP difference comparing 0 vs. 6+ months: −1.03 cm; 95%CI: −1.50 to −0.56 cm; p = 0.001), higher levels of multiple deprivation in childhood (mean height-GaP difference per quintile of

multiple deprivation: −0.22 cm; 95%CI: −0.34 to −0.09 cm; p = 0.001), greater household tobacco smoke exposure during childhood (mean height-GaP difference comparing 0 vs. 20+ hours/week −1.03 cm; 95%CI: −1.64 to −0.42 cm; p < 0.001). Adjusted associations of adult height-GaP with residential outdoor $PM_{2.5}$ exposure in the first year of life (p = 0.593) and "healthy diet" principal component at 3 years old (p = 0.026) did not meet the Bonferroni-corrected threshold of statistical significance. There was no evidence of heterogeneity of height-GaP associations with early-life growth conditions by sex (p-interaction ≥ 0.200). The variance in adult height-GaP explained by a multi-variable regression model including the aforementioned early-life growth conditions was 12.1% (95%CI: 9.8–14.5%).

Unadjusted associations of adult height-GaP with early-life growth conditions and the associations in DMHDS were consistent (Supplementary Fig. 2 *and* Tables 4 and 5).

### Adult height-GaP and later-life mortality

The associations of adult height-GaP with mortality in UKBiobank are summarized in Fig. 2 and Table 1.

In the main adjusted model, a 1-SD deficit in adult height-GaP (−5.2 cm) was associated with higher all-cause mortality (hazard ratio: 1.11; 95%CI: 1.10–1.12). A 1-SD deficit in adult height-GaP was also associated with a higher mortality from atherosclerotic cardiovascular disease (hazard ratio: 1.15; 95%CI: 1.12–1.18) and coronary heart disease (hazard ratio: 1.25; 95%CI: 1.21–1.29). Height-GaP deficit associations with mortality (all-cause, atherosclerotic cardiovascular, and coronary heart disease) were consistent in MESA (Table 1). There was no evidence of heterogeneity by sex- or by race-ethnicity in UKBiobank or MESA (p-interaction ≥ 0.231). Genotype-predicted height was not associated with all-cause or atherosclerotic cardiovascular disease mortality in either UKBiobank or MESA. Measured height —reflecting the linear combination of genotype-predicted height and height-GaP—was associated with both outcomes (Supplementary Table 6).

### Sensitivity analyses

Accounting for potential height-GaP-mortality confounding by later-life height loss using the median and worst-case scenario annualized height loss to "correct" measured height (and thus height-GaP) did not alter the statistical significance of mortality associations and only minimally attenuated the magnitude of association estimates (Supplementary Table 7). Additional adjustment for adult health determinants did not alter statistical significance and minimally attenuated mortality association estimates (Supplementary Table 8). Associations were consistent when restricting the ALSPAC sample to those with non-missing data or those with height measured at the 24 year study visit, and when using ancestry-specific polygenic height scores or computing genotype-predicted height with cohort- and sex-specific models of measured height (Supplementary Tables 9–12).

## Discussion

This study provides robust evidence that the difference between measured and genotype-predicted adult height—here termed "height-GaP"—can serve as a quantitative index of early-life growth adversity and as a predictor of later-life health. In multiple well-characterized cohorts, we demonstrate that adult height-GaP deficit is associated with prospectively ascertained adverse early-life events and exposures—including earlier gestational age at birth, less breastfeeding, greater household tobacco smoke exposure and socioeconomic deprivation—and is also associated with later-life mortality, including death from atherosclerotic cardiovascular disease. These findings support height-GaP as a simple composite index of early-life growth adversity that can be assessed in adulthood, thereby circumventing many of the challenges inherent to the investigation of developmental origins of health and disease.

We observed a clear gradient between adversity during the period of ontogenetic growth and subsequent height-GaP deficit in adulthood in two birth cohorts. These observations are consistent with literature documenting that noxious exposures, suboptimal nutritional environments and socioeconomic disadvantage during fetal and postnatal life attenuate

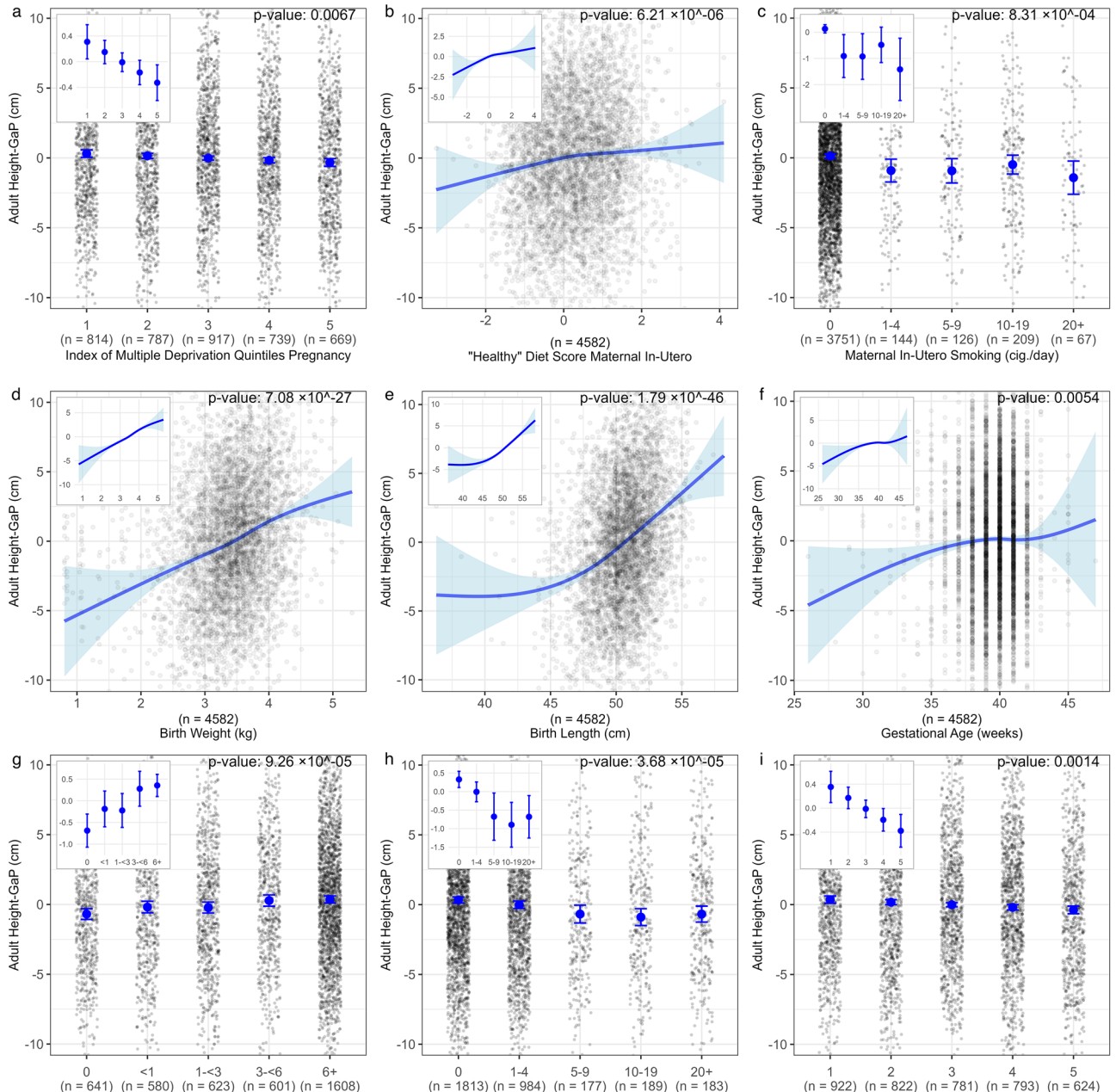

**Fig. 1 | Association of early-life growth conditions with adult height-GaP in ALSPAC.** Estimated marginal mean adult height-GaP associated with **a** maternal English Index of Multiple Deprivation during pregnancy, **b** maternal "healthy" diet score during pregnancy, **c** maternal smoking during pregnancy, **d** birth weight, **e** birth length, **f** gestational age, **g** duration of breastfeeding, **h** environmental tobacco smoke exposure at postnatal ages 6–54 months, and **i** child mean Index of Multiple Deprivation ages 0–12 years. Models are adjusted for sex, age at height-GaP assessment, and genotype-predicted height. Vertical error bars (**a**, **c**, **g**–**i**) and shaded error bands (**b**, **d**–**f**) represent the 95% confidence interval (precision) of the estimated mean height-GaP. Black dots indicate participant data points. *P*-values reflect two-sided Wald tests. Panel insets display the *y*-axis range of the estimated mean and 95%CI. The *p*-value threshold to infer statistical significance was set at 0.050/5 = 0.010 to account for testing of five early-life growth factors (nutrition, peri-natal, noxious exposures, and socio-economic deprivation). ALSPAC Avon Longitudinal Study of Parents and Children, CI confidence interval.

children's realized height[54–56]. The variance in adult height-GaP explained by early-life growth conditions in ALSPAC was 12.1%. Although this fraction may appear modest, it is striking given that we relied on single or intermittent assessments of complex exposures (e.g., diet, neighborhood socio-economic conditions) that likely exert dynamic influences across multiple developmental stages. Height-GaP will facilitate future research seeking to identify the periods of growth susceptibility and emerging threats to healthy human development.

We observed consistent associations between height-GaP and mortality in two adult cohorts including death from atherosclerotic cardiovascular disease—an important observation given that suboptimal early-life growth has long been linked to cardiovascular health in adulthood[16–20,22,24,25,27,31,32]. The lack of association of genotype-predicted height with mortality underscores the potential precision of height-GaP to serve as a surrogate endpoint in early-life intervention trials seeking to improve lifelong health[57].

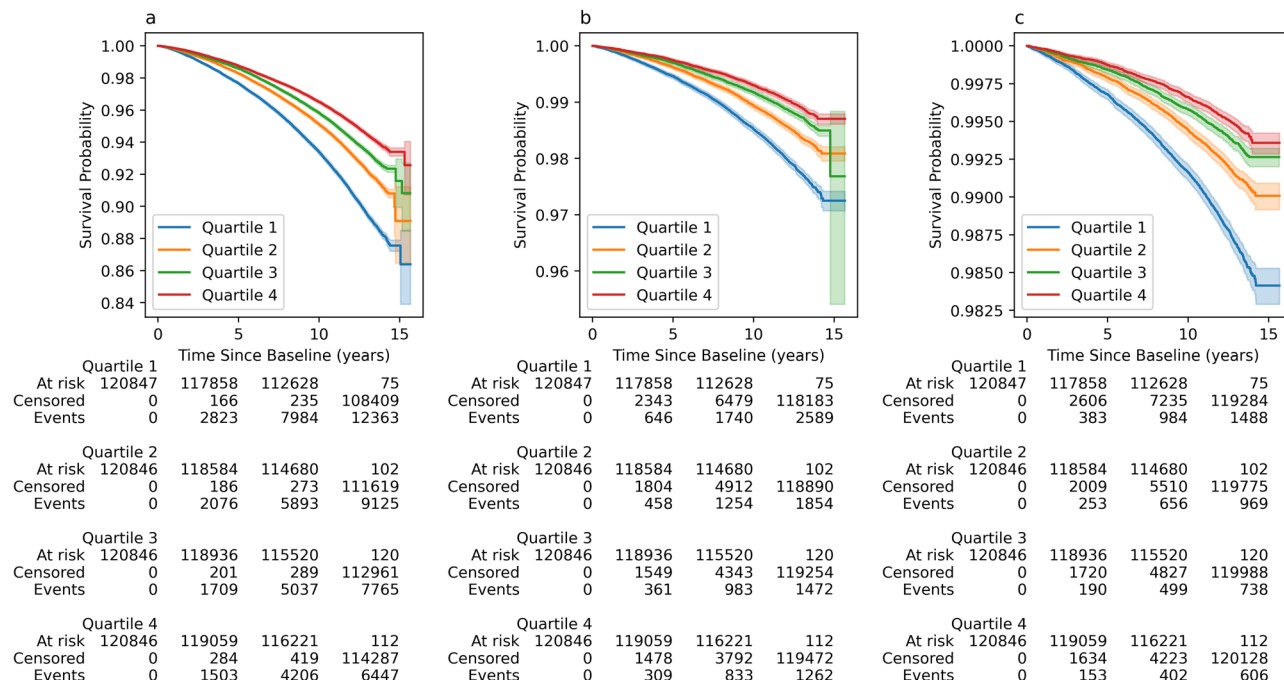

**Fig. 2 | Association of adult height-GaP with mortality.** Kaplan–Meier survival curves by adult height-GaP quartile for **a** all-cause mortality, **b** atherosclerotic cardiovascular disease mortality and **c** coronary heart disease mortality in the UKBiobank. Lower height-GaP quartile represents larger height-GaP deficit. The thresholds to define height-GaP quartile membership were computed for each sex. Shaded error bands represent the 95% confidence interval (precision) of the estimated survival probability.

**Table 1 | Association of height-GaP with mortality in UKBiobank and MESA**

| | Hazard ratio per 1-SD height-GaP deficit (95%CI) p-value | |
|---|---|---|
| | **UKBiobank** | **MESA** |
| **All-cause death** | | |
| Model 1 | 1.11 (1.10, 1.12) $p < 1.00 \times 10^{-100}$ | 1.09 (1.03, 1.15) $p = 4.22 \times 10^{-3}$ |
| Model 2 | 1.11 (1.10, 1.12) $p < 1.00 \times 10^{-100}$ | 1.09 (1.03, 1.15) $p = 4.19 \times 10^{-3}$ |
| **Atherosclerotic cardiovascular disease death** | | |
| Model 1 | 1.15 (1.13, 1.18) $p < 1.00 \times 10^{-100}$ | 1.17 (1.02, 1.34) $p = 2.66 \times 10^{-2}$ |
| Model 2 | 1.15 (1.13, 1.18) $p < 1.00 \times 10^{-100}$ | 1.17 (1.02, 1.34) $p = 2.72 \times 10^{-2}$ |
| **Atherosclerotic coronary heart disease death** | | |
| Model 1 | 1.24 (1.22, 1.28) $p < 1.00 \times 10^{-100}$ | 1.35 (1.14, 1.59) $p = 4.88 \times 10^{-4}$ |
| Model 2 | 1.25 (1.22, 1.29) $p < 1.00 \times 10^{-100}$ | 1.35 (1.14, 1.59) $p = 4.92 \times 10^{-4}$ |

Model 1: age, sex, principal components of genetic ancestry. Model 2: Model 1 variables, genotype-predicted height. *P*-values reported from two-sided Wald tests. The *p*-value threshold to infer statistical significance was set at 0.050/3 = 0.017 to account for testing three mortality outcomes in UKBiobank and *p*-value = 0.050 for replication in MESA.
*MESA* Multi-Ethnic Study of Atherosclerosis.

The biological mechanisms underlying the early-life and later-life height-GaP associations are complex and were not directly interrogated in this study. The same prenatal and early-childhood exposures that impair growth—for instance, poor nutrition, socioeconomic adversity, or exposure to pollutants—may have lasting "programming" effects on immune, metabolic or other homeostatic systems, which in turn, may augment susceptibility to subclinical insults across the life course that culminate in premature mortality. Further mechanistic research should explore whether height-GaP correlates with dysregulation of such systems, thereby elucidating targetable links between early-life adversity and later-life health.

The findings of this study should be interpreted in the context of its limitations. First, the polygenic height score was derived from a sample composed predominantly of European ancestry individuals, which may limit generalizability to other ancestries or regions. We note, however, that the polygenic height score derivation sample included several other ancestries (East Asian: 472,730, Hispanic: 455,180, African: 293,593, South Asian: 77,890)[6]. Moreover, later-life mortality associations were homogeneous across race-ethnic groups in the MESA. Nevertheless, genetic maps of height and validation of height-GaP associations across more diverse ancestries and geographies are needed. Second, GWAS-derived variant effect estimates may reflect both direct and indirect genetic effects. We note, however, that indirect genetic effects included in the polygenic height score would tend to attenuate associations between early-life growth conditions and adult height-GaP. Third, the polygenic height score excludes rare variants. We do not believe this would substantively impact height-GaP applications because (i) rare variants with weak height effects would have weak impact on an individual's height-GaP, and (ii) rare variants with large height effects (e.g., genetic skeletal dysplasias) would be clinically apparent. Fourth, height-GaP associations with later-life health outcomes may reflect residual confounding or mediation by adult health behaviors that correlate

with adverse early-life growth conditions[58]. We believe this is less likely because of the consistency of findings in sensitivity analyses that adjusted for adult risk factors.

This study introduces a precision quantitative index of early-life growth adversity, readily deployable in adulthood, to advance understanding of the developmental origins of disease and facilitate efforts to improve health across the life course.

## Data availability

The data that support the findings of this study are available from each of the respective cohort data coordinating centres, but restrictions apply to the availability of these data, which were used under license for the current study, and so are not publicly available. Data are however available from the corresponding author at benjamin.m.smith@mcgill.ca upon reasonable request and with permission of respective cohort data coordinating centre. Further data access information can be found at the following websites: https://www.bristol.ac.uk/alspac/ https://www.ukbiobank.ac.uk/ http://www.mesa-nhlbi.org https://dunedinstudy.otago.ac.nz/. The source data for Figs. 1, 2 and Supplementary Fig. 2 are provided in the "Supplementary Data 6". Fig. 1 source data is located in the Excel worksheets labelled "Fig. 1 mean and CI" and "Fig. 1 data points", and Fig. 2 source data is in the Excel worksheets labelled "Fig. 2 Kaplan–Meier curves data" and "Fig. 2 Numbers at risk data", Supplementary Fig. 2 source data is in the Excel worksheets labelled "Supplementary Fig. 2 mean and CI" and "Supplementary Fig. 2 data points".

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

## Acknowledgements

This research was supported in part by grants from Canadian Institutes of Health Research. (193740) and the U.S. National Institutes of Health (R01-HL130506, R01-HL077612). The UK Medical Research Council and Wellcome (Grant ref: 217065/Z/19/Z) and the University of Bristol provide core support for ALSPAC. This publication is the work of the authors and will serve as guarantors for the contents of this paper. Genome-wide genotyping data was generated by Sample Logistics and Genotyping Facilities at Wellcome Sanger Institute and LabCorp (Laboratory Corporation of America) using support from 23andMe. A comprehensive list of grants funding and data access policy is available on the ALSPAC website: https://www.bristol.ac.uk/alspac/ We are extremely grateful to all the families who took part in this study, the midwives for their help in recruiting them, and the whole ALSPAC team, which includes interviewers, computer and laboratory technicians, clerical workers, research scientists, volunteers, managers, receptionists and nurses. The UKBiobank data was accessed using the UK Biobank Resource under application number 773778. A comprehensive list of grant funding and data access policy is available on the UKBiobank website: https://www.ukbiobank.ac.uk/ MESA was supported by contracts 75N92020D00001, HHSN268201500003I, N01-HC-95159, 75N92020D00005, N01-HC-95160, 75N92020D00002, N01-HC-95161, 75N92020D00003, N01-HC-95162, 75N92020D00006, N01-HC-95163, 75N92020D00004, N01-HC-95164, 75N92020D00007, N01-HC-95165, N01-HC-95166, N01-HC-95167, N01-HC-95168 and N01-HC-95169 from the National Heart, Lung, and Blood Institute, and by grants UL1-TR-000040, UL1-TR-001079, and UL1-TR-001420 from the National Center for Advancing Translational Sciences (NCATS). Whole genome sequencing (WGS) for the Trans-Omics in Precision Medicine (TOPMed) program was supported by the National Heart, Lung and Blood Institute (NHLBI). WGS for "NHLBI TOPMed: Multi-Ethnic Study of Atherosclerosis (MESA)" (phs001416.v1.p1) was performed at the Broad Institute of MIT and Harvard (3U54HG003067-13S1). Centralized read mapping and genotype calling, along with variant quality metrics and filtering were provided by the TOPMed Informatics Research Center (3R01HL-117626-02S1). Phenotype harmonization, data management, sample-identity QC, and general study coordination, were provided by the TOPMed Data Coordinating Center (3R01HL-120393-02S1). The information contained herein was derived in part from data provided by the Bureau of Vital Statistics, New York City Department of Health and Mental Hygiene. The authors thank the other investigators, the staff, and the participants of the MESA study for their valuable contributions. A full list of participating MESA investigators and institutions and data access policy can be found at http://www.mesa-nhlbi.org. We thank the Dunedin Study members, their families, and friends for their long-term involvement. The Dunedin Multidisciplinary Health and Development Research Unit is based at University of Otago within the Ngāi Tahu tribal area whom we acknowledge as first peoples, tangata whenua (transl. people of this land). We thank Dunedin Unit research staff, previous Study Director, Emeritus Distinguished

Professor, the late Richie Poulton, for his leadership during the Study's research transition from young adulthood to aging (2000-2023) and Study founder, Dr Phil A. Silva. The Dunedin Multidisciplinary Health and Development Research Unit is supported by the New Zealand Health Research Council (Programme Grant 16-604) and has also received funding from the New Zealand Ministry of Business, Innovation and Employment. The study is also supported by the US-National Institute of Aging Grant R01AG032282, and UK Medical Research Council Grant MR/X021149/1. The data access policy can be found at https://dunedinstudy.otago.ac.nz/.

## Author contributions

Conception or design of the work: R.G.-P., C.L.D., A.M., J.D., R.J.H., B.M.S. Acquisition, analysis, or interpretation of data: R.G.-P., M.P.A., R.B., SEC, C.L.D., J.P.A., A.A., A.G.B., A.C., S.L.-D., M.P.E., J.C.E., D.R.J., D.M., A.M., E.D.M., T.E.M., E.C.O., S.R., S.S.R., C.S., S.S., P.S., K.S., R.T., K.E.W., B.W., B.Y., J.D., S.O.S., R.G.B., R.J.H., B.M.S. Drafted the work or substantively revised it: R.G.-P., A.C., D.M., A.M., T.E.M., R.J.H., B.M.S. Reviewed and approved the submitted version: all authors.

## Competing interests

The authors declare no competing interests.

## Additional information

Raphael Goldman-Pham [1], Matthew P. Alter[1], Rebecca Bao[1], Sophie É. Collins [1], Catherine L. Debban[2], James P. Allinson[3], Antony Ambler[4], Alain G. Bertoni[5], Avshalom Caspi [6,7], Stephanie Lovinsky-Desir[8], Magnus P. Ekstrom[9], James C. Engert [1], David R. Jacobs Jr [10], Daniel Malinsky[11], Ani Manichaikul [2], Erin D. Michos [12], Terrie E. Moffitt[6,7], Elizabeth C. Oelsner[13], Sandhya Ramrakha[4], Stephen S. Rich [2], Coralynn Sack[14], Sanja Stanojevic[15], Padmaja Subbarao [16], Karen Sugden[6], Reremoana Theodore[4], Karol E. Watson[17], Benjamin Williams[6], Bin Yang[11], Josée Dupuis [18], Seif O. Shaheen[19,20], R. Graham Barr[13], Robert J. Hancox [21] & Benjamin M. Smith [1,13,18] ✉

[1]Department of Medicine, Faculty of Medicine and Health Sciences, McGill University, Montreal, QC, Canada. [2]Department of Genome Sciences, University of Virginia, Charlottesville, VA, USA. [3]Faculty of Medicine, Imperial College London National Heart and Lung Institute, London, UK. [4]Department of Psychology, University of Otago - Ōtākou Whakaihu Waka, Dunedin, New Zealand. [5]Department of Public Health Sciences, Wake Forest University, Winston-Salem, NC, USA. [6]Department of Psychology and Neuroscience, Duke University, Durham, NC, USA. [7]Institute of Psychiatry, Psychology and Neuroscience, Kings College London, London, UK. [8]Department of Pediatrics, College of Physicians and Surgeons, Columbia University, New York, NY, USA. [9]Department of Clinical Sciences, Lund University, Lund, Sweden. [10]School of Public Health, University of Minnesota, Minneapolis, MN, USA. [11]Department of Biostatistics, Mailman School of Public Health, Columbia University, New York, NY, USA. [12]Department of Medicine, School of Medicine, Johns Hopkins University, Baltimore, MD, USA. [13]Department of Medicine, College of Physicians and Surgeons, Columbia University, New York, NY, USA. [14]Department of Medicine and Department of Environmental & Occupational Health Sciences, School of Medicine, University of Washington, Seattle, WA, USA. [15]Department of Community Health and Epidemiology, Faculty of Medicine, Dalhousie University, Halifax, NS, Canada. [16]Department of Paediatrics and Dalla Lana School of Public Health, University of Toronto, Toronto, Canada. [17]Department of Medicine, David Geffen School of Medicine, University of California Los Angeles, Los Angeles, CA, USA. [18]Department of Epidemiology, Biostatistics and Occupational Health, School of Population and Global Health, Faculty of Medicine and Health Sciences, McGill University, Montreal, QC, Canada. [19]Wolfson Institute of Population Health, Barts and The London School of Medicine and Dentistry, Queen Mary University of London, London, UK. [20]Allergy and Lung Health Unit, Melbourne School of Population and Global Health, The University of Melbourne, Melbourne, VIC, Australia. [21]Department of Preventive & Social Medicine, Dunedin School of Medicine, University of Otago - Ōtākou Whakaihu Waka, Dunedin, New Zealand. ✉e-mail: benjamin.m.smith@mcgill.ca; bs2723@cumc.columbia.edu

