## [Transparent Peer Review file · Communications Medicine]

Quantifying the impact of early-life growth adversity on later-life health

Corresponding Author: Dr Benjamin Smith

Version 0:

Reviewer comments:

Reviewer #1

(Remarks to the Author)

This study by Goldman-Pharm et al. examined the relationship between the discrepancy of genetically predicted height and measured height with early-life predictors in the ALSPAC cohort, as well as with mortality outcomes in the UK Biobank. Importantly, the authors successfully replicated their findings in two independent cohorts, strengthening the robustness of the results. The research question addressed is timely and of significant interest, and the manuscript is clearly and thoughtfully written. The authors demonstrate that the genetic height-actual height gap (height-GaP) is meaningfully associated with early-life factors and later health outcomes. Overall, I find the work to be a valuable contribution to the field. I have only a few comments.

Major

1. ALSPAC included height measurement from either 18years or 24years of age, I would suggest a sensitivity analysis using data with measured height at age 24years only, as not also the total attained height, but also the timing of growth might be affected and not all the participants might have reached their maximum height already at age 18 years
2. In my opinion, sex interactions should be tested additionally or sensitivity analyses stratified by sex need to be performed.

Minor

3. Please report shortly about the results of the “complete data” analyses
4. Page 4 methods: typo “ALPAC”
5. Page 5 Genotype-predicted adult height: Please clarify what is meant by “sample effect allele prevalence”, the effect allele frequencies in Yengo et al. or obtained from a reference population?
6. Page 6 statistical methods: How was “whether the linear or spline model fit was superior” ascertained
7. Page 7: Please provide details on used package(s) for imputation of missing data
8. Page 9 results: “Genotype-predicted height was not associated with higher all-cause or atherosclerotic cardiovascular disease mortality in either UKBiobank or MESA.” – remove “higher”
9. Pages 19&20 Table 1&2:
 - a. I would suggest to add p-values for the difference of each of the variables between the height-GAP Quartiles
 - b. Additionally, I think all height variables should be presented for males and females separately
 - c. Were height-GAP Quartiles calculated sex-specific or using the total population’s cut offs?
 - d. “height” in table 1 vs. “measured height” in table 2
10. Supplemental Figure 1: Please clarify what is meant by “Later-life health factors” and why these are related to Early-life growth conditions, additionally I was wondering why a causal relationship between adult height and later-life health outcomes cannot be hypothesized.
11. Supplemental Table 2&3: Height as well as Genotype predicted height seems to be available for Excluded participants, please also add the % missing values or provide a flow chart of the study population and the missingness of information

Reviewer #2

(Remarks to the Author)

Thank you for the interesting manuscript. The authors assess using ‘height-GaP’ – the difference between SNP-predicted height and adult obtained height – as an indicator of early life growth adversity and its association with later-life mortality.

This is an important topic particularly within the life course framework and an indicator summarising early life adversity would be valuable addition to many studies, especially when no information on early life is available or it is very limited. In the setting of the current manuscript, the requirement for genetic information would limit its accessibility in future studies.

In the current study, they found that several maternal, birth and early life factors with established risks for later life health were associated with height-GaP. Also, later life all-cause mortality and atherosclerotic cardiovascular disease mortality seemed to be associated with this estimate.

The rationale behind the study is intuitive and clear. Methodology is well thought through, and the authors provide additional analyses and information on the samples to support the main analyses and conclusions.

Below, there are a few comments I would like to raise.

1. The rationale of the study is introduced clearly and logically, but references to the current state of literature in the field are lacking. Do any previous attempts to summarise or estimate early life growth adversities exist? More broadly, the use of references in the introduction is limited: there are seven citations altogether, five of which refer to the association between ASCVD and early life adversity (and are 25-40 years old).
2. In the lines 225-227 the PCs in different cohorts are described. I found that the mortality analyses were adjusted for PCs. Were they also adjusted for in some analysis using DMHDS? If yes, it would be good to explicitly say that with ALSPAC PCs were not adjusted for if that was the case.
3. I was wondering why health insurance status was used as a covariate in MESA in addition to family income and educational attainment? In the manuscript in lines 227-230 it sounds like this variable was also involved in UKB.
4. Lines 251-254: does the P-value threshold in DMHDS refer to the association analyses or imputation? What predictors were used in the imputation process for different variables?
5. Lines 257-260: Extended model 2 was adjusted for, among other variables, PCs, and a product term for sex by race-ethnicity was included. Was race-ethnicity also included as a main effect, and was there collinearity with the PCs? I did not find the results of these analyses in the provided material.
6. In Supplemental Table 3, BMI -Missing , Hypertension, Lipid lowering medication, Physical activity lines are empty. I am not sure what values are in the line with 'Ethnicity, no. (%)'
7. I find that the title could be more specific by including the main concept of the paper, using the contrast of attained vs SNP-predicted height to quantify early life growth adversities.
8. The manuscript focuses on the deficit in attained adult height compared to the SNP-predicted height. How would the authors interpret the other end of the height-GaP scale, i.e., those who grew taller than their SNP-predicted height was? Related to this, in Table 2 - Ethnicity. Over 70% of UKB participants with ethnicity being 'black' are in the fourth quartile of Height GaP. In MESA the participants were more equally distributed in the different quartiles. What is the author's interpretation of this, is this likely due to the original sample where the polygenic height score was derived, the imputation panel or something else?
9. The methods section of the abstract (lines 79-80) describes variables that were used to adjust the regression models, but it does not mention what were the independent and dependent variables in these regression models.
10. In Table 1, Age at height-GaP assessment: IQR is mentioned in the variable description but not 'median'
11. Figure 2: Please explain the N's below the figures. I assume they are from the timepoints (0, 5, 10, 15 yrs) marked in the figure but this is not clear.

Reviewer #3

(Remarks to the Author)

The manuscript is clear and well-organized, and it presents a relevant contribution to the field. A particular strength of the study is that the authors test their hypothesis in multiple independent cohorts, which supports the robustness of the findings. I have a few minor remarks that may help to further clarify certain points.

- Lines 143-145: What's the difference between initial recruitment and further recruitment?
- Lines 159-162: How many were initially recruited?
- Lines 193-194: If I understand correctly, a lower height-GaP means that the measured adult height is close to the genotype-predicted height? Hence, a smaller deficit?
- Lines 204-206: Is this a measure for socio-economic status? What do the authors mean by multiple deprivation during pregnancy?

Version 1:

Reviewer comments:

Reviewer #1

(Remarks to the Author)

The authors have addressed my comments thoroughly and to my full satisfaction. The revisions have notably improved the clarity and overall quality of the manuscript. The statistical analyses are appropriate and valid, and all necessary information to reproduce the work has been provided. I consider the paper suitable for publication in its current form.

Reviewer #2

(Remarks to the Author)

I would like to thank the authors for their careful revision. I am satisfied with the way they have addressed my comments and concerns.

Reviewer #3

(Remarks to the Author)

The authors have addressed my previous comments, and the manuscript has improved considerably.

July 31, 2025

We thank the Editors and Reviewers for their constructive feedback. Please find our responses below with corresponding edits to the manuscript.

R1.1. ALSPAC included height measurement from either 18 years or 24 years of age, I would suggest a sensitivity analysis using data with measured height at age 24 years only, as not also the total attained height, but also the timing of growth might be affected and not all the participants might have reached their maximum height already at age 18 years

A sensitivity analysis using the subset of participants with height measured at the 24 year study visit has been added to the manuscript. Beta estimates are consistent in this sample.

Methods: "Finally, ALSPAC analyses were repeated among participants ... with height measured at the 24 year study visit ..."

Results: "Associations were consistent when restricting the ALSPAC sample to those with ... height measured at the 24 year study visit ... (Supplemental Table 13)."

Supplementary Table 13: Association of early-life growth conditions with adult height-GaP in ALSPAC restricted to participants with height assessed at the 24 year study visit.

	Adjusted mean difference in height-GaP (95%CI) p-value
Pregnancy index of multiple deprivation, per 1-quintile increment	-0.13 (-0.27, 0.08) p=0.069
Maternal smoking during pregnancy	p=0.014
Never	Reference
1-4 cigarettes/day, n (%)	-0.74 (-1.77, 0.28)
5-9 cigarettes/day, n (%)	-1.20 (-2.29, -0.11)
10-19 cigarettes/day, n (%)	-0.91 (-1.77, -0.04)
20+ cigarettes/day, n (%)	-1.34 (-2.87, 0.20)
"Healthy" diet principal component during pregnancy, per 1-SD decrement	-0.50 (-0.70, -0.30) p<0.001
Gestational age at birth, weeks, no. (%)	P=0.009
38+	Reference
32-<38	-0.25 (-0.87, 0.37)
<32	-5.93 (-10.86, -0.99)
Birth length, cm, per 1-SD decrement	-0.68 (-0.82, -0.54) p<0.001
Birth weight, kg, per 1-SD decrement	-2.34 (-2.83, -1.85) p<0.001
Breastfeeding duration	P=0.014
6+ months	Reference
3-<6 months	-0.11 (-0.66, 0.45)
1-<3 months	-0.43 (-0.99, 0.13)
<1 month	-0.41 (-1.00, 0.19)
Never	-0.96 (-1.52, -0.39)
Childhood (ages 0-12 years) index of multiple deprivation, per 1-quintile increment	-0.13 (-0.26, 0.01) p=0.063
Childhood (ages 6-34 months) household tobacco smoke exposure duration	p<0.001
0 hours/week	Reference
>0-5 hours/week	-0.39 (-0.80, 0.03)
>5-10 hours/week	-1.27 (-2.12, -0.41)
>10-20 hours/week	-1.26 (-2.07, -0.46)
20+ hours/week	-1.20 (-1.98, -0.42)
"Healthy" diet principal component at 38 months, per 1-SD decrement	-0.19 (-0.40, 0.03,) p=0.086
Residential outdoor PM _{2.5} concentration, µg/m ³ , per 1-SD decrement	0.10 (-0.14, 0.34) p=0.421

Model covariables: age at height-GaP assessment, sex and genotype-predicted height.
 Abbreviations: ALSPAC = Avon Longitudinal Study of Parents and Children; SD = standard deviation;
 PM_{2.5} = particulate matter with diameter less the 2.5 micrometres.

R1.2. In my opinion, sex interactions should be tested additionally or sensitivity analyses stratified by sex need to be performed.

A secondary analysis evaluating sex interactions by early-life adversity has been added to the manuscript. There was no evidence of interaction by sex. [Sex interaction tests for the adult mortality analyses were previously included in the original version]. A full table of interactions between early-life exposures and sex is provided below:

Independent variable x Sex	Interaction P-Value
Pregnancy index of multiple deprivation, per 1-quintile increment	0.602
“Healthy” diet principal component during pregnancy, per 1-SD decrement	0.627
Maternal smoking during pregnancy	0.710
Birth weight, kg, per 1-SD decrement	0.823
Birth length, cm, per 1-SD decrement	0.537
Gestational age at birth, weeks	0.255
Breastfeeding duration	0.772
“Healthy” diet principal component at 38 months, per 1-SD decrement	0.363
Childhood (ages 6-34 months) household tobacco smoke exposure duration	0.968
Childhood (ages 0-12 years) index of multiple deprivation, per 1-quintile increment	0.718
Residential outdoor PM _{2.5} concentration, µg/m ³ , per 1-SD decrement	0.200

The manuscript has been modified to include this analysis:

Methods: “Heterogeneity of associations by sex were evaluated by adding a height-GaP-sex product term to each regression model.”

Results: “There was no evidence of heterogeneity of height-GaP associations with early-life growth conditions by sex (p-interaction ≥ 0.200).”

R.1.3. Please report shortly about the results of the “complete data” analyses

The results are included in the manuscript.

Results: “Associations were consistent when restricting the ALSPAC sample to those with non-missing data ... (Supplemental Table 12).

Supplementary Table 12: Association of early-life growth conditions with adult height-GaP in ALSPAC restricted to participants with non-missing data.

	Adjusted mean difference in height-GaP (95%CI) p-value
Pregnancy index of multiple deprivation, per 1-quintile increment	-0.19 (-0.30, -0.08) p=0.001
Maternal smoking during pregnancy	p<0.001
Never	Reference
1-4 cigarettes/day, n (%)	-1.07 (-1.90, -0.23)
5-9 cigarettes/day, n (%)	-1.07 (-1.96, -0.18)
10-19 cigarettes/day, n (%)	-0.68 (-1.38, 0.02)
20+ cigarettes/day, n (%)	-1.71 (-2.92, -0.50)

"Healthy" diet principal component during pregnancy, per 1-SD decrement	-0.43 (-0.59, -0.27) p<0.001
Gestational age at birth, weeks, no. (%)	p<0.001
38+	Reference
32-<38	-0.07 (-0.58, 0.43)
<32	-5.37 (-7.23, -3.50)
Birth length, cm, per 1-SD decrement	-0.73 (-0.82, -0.65) p<0.001
Birth weight, kg, per 1-SD decrement	-2.41 (-2.68, -2.13) p<0.001
Breastfeeding duration	p<0.001
6+ months	Reference
3-<6 months	-0.07 (-0.54, 0.40)
1-<3 months	-0.60 (-1.06, -0.14)
<1 month	-0.56 (-1.04, -0.09)
Never	-1.07 (-1.53, -0.62)
Childhood (ages 0-12 years) index of multiple deprivation, per 1-quintile increment	-0.23 (-0.36, -0.11) p<0.001
Childhood (ages 6-34 months) household tobacco smoke exposure duration	p<0.001
0 hours/week	Reference
>0-5 hours/week	-0.28 (-0.67, 0.11)
>5-10 hours/week	-1.50 (-2.27, -0.73)
>10-20 hours/week	-1.30 (-2.05, -0.55)
20+ hours/week	-0.95 (-1.71, -0.19)
"Healthy" diet principal component at 38 months, per 1-SD decrement	-0.23 (-0.39, -0.07) p=0.005
Residential outdoor PM _{2.5} concentration, µg/m ³ , per 1-SD decrement	0.04 (-0.16, 0.24) p=0.704

Model covariables: age at height-GaP assessment, sex and genotype-predicted height.

Abbreviations: ALSPAC = Avon Longitudinal Study of Parents and Children; SD = standard deviation; PM_{2.5} = particulate matter with diameter less than 2.5 micrometres.

R1.4. Page 4 methods: typo "ALPAC"

Typo corrected.

R1.5. Page 5 Genotype-predicted adult height: Please clarify what is meant by "sample effect allele prevalence", the effect allele frequencies in Yengo et al. or obtained from a reference population?

Apologies for this ambiguity. For missing effect alleles we assigned the value of the effect allele frequency of the participant's respective cohort. The Methods text has been revised.

Methods: "missing effect alleles were assigned the effect allele frequency of the participant's respective cohort"

R1.6. Page 6 statistical methods: How was "whether the linear or spline model fit was superior" ascertained

Apologies for the lack of clarity. Model superiority was based on Akaike information criterion. The Methods text has been revised.

Methods: "Tabular results are reported per 1-SD or quantile contrast depending on whether the linear or spline model fit was superior, as assessed according to model Akaike Information Criterion."

R1.7. Page 7: Please provide details on used package(s) for imputation of missing data. Additional details have been added to the Methods section.

Methods: "Imputation of missing data using all analysis variables as predictors in ALSPAC and DMHDS was performed with polytomous logistic regression and predictive mean matching for

categorical and continuous variables, respectively, across 100 imputed datasets ('MICE' R package, v 3.16.0)."

R1.8. Page 9 results: "Genotype-predicted height was not associated with higher all-cause or atherosclerotic cardiovascular disease mortality in either UKBiobank or MESA."
– remove "higher".

Text revised accordingly.

R1.9. Pages 19&20 Table 1&2:

a) I would suggest to add p-values for the difference of each of the variables between the height-GAP Quartiles

P-values have been added to the participant characteristic tables, though we note that these tables are intended to be descriptive rather than the basis of statistical inference. The arbitrary number of rows presented (multiple testing), co-linearity and sample size may influence the number of 'significant' p-values and hinder valid statistical inferences.

b) Additionally, I think all height variables should be presented for males and females separately

Done.

c) Were height-GaP Quartiles calculated sex-specific or using the total population's cut offs?

The height-GaP quartile thresholds were sex-specific. This important detail has been added to the Table legends.

d) "height" in table 1 vs. "measured height" in table 2

Apologies for this ambiguity. Both tables report measured height. The row header has been edited accordingly.

R1.10. Supplemental Figure 1: Please clarify what is meant by "Later-life health factors" and why these are related to Early-life growth conditions, additionally I was wondering why a causal relationship between adult height and later-life health outcomes cannot be hypothesized.

We apologize for this ambiguity: "Later-life health factors" refers to later-life exposures/events/behaviours (e.g., adult tobacco smoking, adult obesity, adult hypertension, etc...) that are causally or potentially causally related to later-life health outcomes (e.g. mortality). In our evaluation of the relationship between height-GaP and later-life mortality, some members of our writing group noted that some "Early-life growth conditions" are associated with "Later-life health factors" (e.g., pre-term birth and adult hypertension) and were interested in whether height-GaP-mortality associations were independent of these later-life health factors (in part because these later-life factors can be readily assessed in a clinic or research setting [e.g., smoking behaviours, body mass index, blood pressure]). Supplemental Figure 1 sought to acknowledge this causal (or potentially causal) pathway and the manuscript included a sensitivity analysis in which height-GaP mortality models were adjusted for "Later-life health factors" (Supplement Table 11). The Supplemental Figure 1 legend has been revised and expanded to improve clarity.

Adult height has been correlated with later-life health outcomes in several epidemiologic studies. (e.g., ^{1,2}) Our manuscript introduces a quantitative index to facilitate investigations into the early-life causal pathways and later-life implications of this height-health correlation. We therefore do not include a causal edge between adult height and later-life health outcomes.

Supplement: Supplemental Figure 1 Legend: “A directed acyclic graph depicting hypothesized causal relationships of early-life growth conditions with late-life health outcomes. “Later-life health factors” refers to later-life exposures/events/behaviours (e.g., adult hypertension) that may result from early-life growth adversity (e.g., preterm) and are potentially causally related to later-life health outcomes (e.g. mortality).”

R1.11. Supplemental Table 2&3: Height as well as Genotype predicted height seems to be available for Excluded participants, please also add the % missing values or provide a flow chart of the study population and the missingness of information

Done.

Supplemental Table 2 (excerpt): Characteristics of ALSPAC participants included and excluded from analyses.

	Included	Excluded
No.	4,582	11,063
Genotype-predicted height cm	171.9 (7.9)	173.8 (7.7) [n=4215, % missing=61.9]
Height-GaP, cm	0.0 (5.1)	NA [n=0, % missing=100.0]
Height, cm	171.9 (9.4)	170.5 (9.1) [n=1414, % missing=87.2]

Supplemental Table 3 (excerpt): Characteristics of participants in the UKBiobank included and excluded from the analysis.

	Included	Excluded
No.	483,385	18,994
Genotype-predicted height cm	168.5 (7.7)	169.1 (7.7) [n=3,777, % missing=80.2]
Height-GaP, cm	0.0 (5.2)	-2.3 (5.1) [n=2,339, % missing=87.7]
Measured height, cm	168.5 (9.3)	167.1 (9.2) [n=16,453, % missing=13.4]

R2.1. The rationale of the study is introduced clearly and logically, but references to the current state of literature in the field are lacking. Do any previous attempts to summarise or estimate early life growth adversities exist? More broadly, the use of references in the introduction is limited: there are seven citations altogether, five of which refer to the association between ASCVD and early life adversity (and are 25-40 years old).

Thank you for this comment and we agree, the evidence supporting the importance of early-life events and exposures to later-life health is extensive. We have added 27 citations to the introduction and text acknowledging existing early-life growth adversity estimation methods.

Introduction: “... Anthropometric indices of early-life growth adversity readily measured in adulthood—such as measured height, sitting-to-standing-height ratio and leg-to-torso length ratio³⁻⁶—are determined in part by genetics,⁷⁻⁹ reducing their utility as a quantitative marker of non-genetic early-life growth adversity. Molecular indices of early-life growth adversity such as adult telomere length and DNA methylation signatures (e.g., epigenetic age) are limited by their continued plasticity in adulthood.¹⁰⁻¹⁶ ...

... The second objective was to determine whether adult height-GaP is associated with later-life mortality from all-causes and from atherosclerotic cardiovascular disease, the latter being a major public health burden with established developmental origins.¹⁷⁻³⁵”

R2.2. In the lines 225-227 the PCs in different cohorts are described. I found that the mortality analyses were adjusted for PCs. Were they also adjusted for in some analysis using DMHDS? If yes, it would be good to explicitly say that with ALSPAC PCs were not adjusted for if that was the case.

Apologies for this ambiguity. The PCs were not adjusted for in the DMHDS for consistency with the ALSPAC study analysis (which had already stratified according to PCs). The text has been revised for clarity. The DMHD associations are consistent with additional adjustment for PC1-5.

Methods:

“Early-life associations were not further adjusted for principal components of genetic ancestry due to the ALSPAC cohort’s prior sampling according to principal components.”

...

“The threshold for association analyses in the DMHDS replication sample, which was similarly not adjusted for principal components of genetic ancestry, was a two-sided p-value=0.050.”

R2.3. I was wondering why health insurance status was used as a covariate in MESA in addition to family income and educational attainment? In the manuscript in lines 227-230 it sounds like this variable was also involved in UKB.

MESA is a U.S.-based cohort where adult healthcare access is market-based, multi-payer and not “universal”. In contrast, UK has a single-payer “universal” healthcare system. In the sensitivity analysis that sought to account for adult health-related factors that may be related to both early-life adversity and later-life health (Supplemental Table 11), we adjusted for adult health insurance status in MESA. The Methods text and Table legend have been revised to specify that health insurance status was adjusted for in MESA only.

Methods: “... health insurance status (in MESA only) were derived from standardized questionnaire items.”

Supplemental Table 11 Legend: “Model covariables: age, sex, principal components of genetic ancestry, genotype-predicted height, cigarette smoking status, pack-years, alcohol use status, drinks per week, minutes of moderate and of vigorous physical activity per week, weight class, diabetes status, hypertension status, systolic blood pressure, LDL cholesterol concentration, cholesterol-lowering medication use, educational attainment, health insurance status (in MESA) and household income.”

R2.4. Lines 251-254: does the P-value threshold in DMHDS refer to the association analyses or imputation? What predictors were used in the imputation process for different variables?

Apologies for the lack of clarity. The p-values threshold refers to the association analyses. DMHDS multiple imputation was performed using all analysis variables as predictors. The text has been revised for clarity.

Methods: “...the threshold for association analyses in the DMHDS replication sample was a two-sided p-value=0.050.”

“Imputation of missing data using all analysis variables as predictors in ALSPAC and DMHDS was performed with polytomous logistic regression and predictive mean matching for categorical and continuous variables, respectively, across 100 imputed datasets (‘MICE’ R package, v 3.16.0).”

R2.5. Lines 257-260: Extended model 2 was adjusted for, among other variables, PCs, and a product term for sex by race-ethnicity was included. Was race-ethnicity also included as a main effect, and was there collinearity with the PCs? I did not find the results of these analyses in the provided material.

*Apologies for the lack of specificity. Genetic ancestry PCs were omitted from the race-ethnicity interaction models due to concerns about collinearity. A race-ethnicity main effect term was included in the model, in addition to height-GaP*race-ethnicity product term:*

*Mortality = age, age² + sex + genotype-predicted height + height-GaP + race-ethnicity + height-GaP * race-ethnicity.*

The text has been revised to improve clarity.

Methods: “Heterogeneity of associations by sex and by race-ethnicity were evaluated by including main effect and product terms in regression models; the race-ethnicity interaction models excluded principal components of genetic ancestry.”

Incidentally, retaining the ancestry PCs in the model yielded near-identical results:

Outcome	Minimum p-value for height-GaP*race-ethnicity term	
	Ancestry PCs omitted from model (original analysis)	Ancestry PCs added to model
All-cause mortality	0.61	0.63
CVD mortality	0.48	0.57
CHD mortality	0.85	0.93

*Proportional hazards mortality model: age, age², sex, genotype-predicted height, height-GaP, race-ethnicity, height-GaP*race-ethnicity, without and with PCs included.*

R2.6. In Supplemental Table 3, BMI -Missing , Hypertension, Lipid lowering medication, Physical activity lines are empty. I am not sure what values are in the line with ‘Ethnicity, no. (%)’

Apologies for this oversight. The missing data have been added to the Table. The description of the “Ethnicity, no. (%)” row has been revised to “Self-reported ethnicity, no. of participants (%)”

R2.7. I find that the title could be more specific by including the main concept of the paper, using the contrast of attained vs SNP-predicted height to quantify early life growth adversities.

Proposed title revision: “Quantifying the impact of early-life growth adversity on later-life health using the difference between attained and genotype-predicted adult height”

R2.8. The manuscript focuses on the deficit in attained adult height compared to the SNP-predicted height. How would the authors interpret the other end of the height-GaP scale, i.e., those who grew taller than their SNP-predicted height was? Related to this, in Table 2 - Ethnicity. Over 70% of UKB participants with ethnicity being ‘black’ are in the fourth quartile of Height GaP. In MESA the participants were more equally distributed in the different quartiles. What is the author’s interpretation of this, is this likely due to the original sample where the polygenic height score was derived, the imputation panel or something else?

Genotype-predicted height values were obtained by fitting linear a regression model of measured adult height using the polygenic height scores. This method results in an approximate 50% split of residuals above and below the genotype-predicted values when the residuals are symmetrically distributed, as was the case for our datasets. The ‘sign’ of a participant’s height-GaP (i.e. positive vs. negative) is thus a consequence of regression-based approach to computing genotype-predicted height rather than an indication that someone ‘surpassed’ their genetic height potential. Height-GaP is best interpreted as a continuous index, whereby smaller values are associated with greater early-life growth adversity compared to larger values.

We had considered a linear translation of height-GaP values, whereby the largest height-GaP value was assigned a value of 0 cm (i.e., ‘zero’ growth deficit) resulting in all other values being negative (i.e., some growth deficit). However, this added step would result in expressing all height-GaP values relative to an outlier and, in preliminary presentations, tended to introduce confusion to audiences.

Regarding the differences in race-ethnic proportions by height-GaP quartile in UKBiobank and MESA: Thank you for detecting and highlighting this critical oversight. The cause has been identified and is a computational inconsistency resulting from having different analysts

independently perform the primary (UKB) and replication (MESA) analyses (with the intended goal increasing rigor). Briefly, in UKBiobank genotype-predicted height was calculated by regressing measured height on all-ancestry polygenic height score stratified by sex. In MESA, however, genotype-predicted height was calculated by regressing measured height on the all-ancestry polygenic height score stratified by sex and by race-ethnicity. This was done to account for apparent population stratification of polygenic scores in MESA that was evident when examining the height-by-polygenic height scores by race ethnicity (**Figure R1**). Differences in all-ancestry polygenic risk score distributions by race-ethnic group were not apparent in UKBiobank due extreme imbalance in race-ethnic proportions (White: 94.3%, Black: 1.6%, Black: 1.6%, Others: 1.5%).

When the UKBiobank genotype-predicted height calculation is harmonized to MESA by regressing measured height on all-ancestry polygenic height score by sex and race-ethnic strata, the race-ethnic proportions are balanced across height-GaP quartile, as seen in the MESA data. Conversely, when the MESA genotype-predicted height is calculated by regressing measured height on polygenic height score stratified by sex alone, the differences in race-ethnic proportions by height-GaP quartile are evident.

Critically, height-GaP associations with mortality are consistent regardless of the method used to compute genotype predicted height.

We have updated the Methods and Results to include both approaches to calculating genotype-predicted height (Tables 2 and 3, Supplemental Tables 5 and 15).

Figure R1. Scatter plots of measured height vs. all-ancestry derived polygenic height score among males (left) and females (right) in MESA highlighting the differences in polygenic height score distributions by race-ethnic group with preserved correlations between measured height and polygenic height score. Note: y-axis range differs for male and female panels.

Table 2 (excerpt): Characteristics of participants included in the UKBiobank analyses.

	All	By Height-GaP Quartile			
		1	2	3	4
Self-reported ethnicity, no. of participants (%)					
White	455,857 (94.3)	113,088 (93.6)	114,597 (94.8)	114,853 (95.0)	113,319 (93.8)
Black	7,511 (1.6)	2,222 (1.8)	1,589 (1.3)	1,589 (1.3)	2,111 (1.7)
South Asian	9,175 (1.9)	2,453 (2.0)	2,203 (1.8)	2,133 (1.8)	2,386 (2.0)
Chinese	1,491 (0.3)	369 (0.3)	397 (0.3)	334 (0.3)	391 (0.3)

Other	4,279 (0.9)	895 (0.7)	804 (0.7)	873 (0.7)	1,707 (1.4)
Do not know	200 (0.0)	57 (0.0)	44 (0.0)	41 (0.0)	58 (0.0)
Prefer not to answer	1,562 (0.3)	392 (0.3)	381 (0.3)	361 (0.3)	428 (0.4)

Supplementary Table 5 (excerpt): Characteristics of participants included in the Multi-Ethnic Study of Atherosclerosis.

	All	By Height-GaP Quartile			
		1	2	3	4
Self-reported race-ethnicity, no. (%)					
White	2,486 (39.1)	576 (36.3)	657 (41.4)	668 (42.1)	585 (36.8)
Black	1,660 (26.1)	464 (29.2)	379 (23.9)	363 (22.9)	454 (28.6)
Hispanic	1,437 (22.6)	384 (24.2)	336 (21.2)	340 (21.4)	377 (23.7)
Chinese	769 (12.1)	164 (10.3)	216 (13.6)	217 (13.7)	172 (10.8)

Table 3: Association of height-GaP with mortality in UKBiobank and MESA.

	Hazard ratio per 1-SD height-GaP deficit (95%CI) p-value	
	UKBiobank	MESA
All-cause death		
Model 1	1.11 (1.10, 1.12) p<0.001	1.09 (1.03, 1.15) p=0.004
Model 2	1.11 (1.10, 1.12) p<0.001	1.09 (1.03, 1.15) p=0.004
Atherosclerotic cardiovascular disease death		
Model 1	1.15 (1.13, 1.18) p<0.001	1.17 (1.02, 1.34) p=0.027
Model 2	1.15 (1.13, 1.18) p<0.001	1.17 (1.02, 1.34) p=0.027
Atherosclerotic coronary heart disease death		
Model 1	1.24 (1.22, 1.28) p<0.001	1.35 (1.14, 1.59) p<0.001
Model 2	1.25 (1.22, 1.29) p<0.001	1.35 (1.14, 1.59) p<0.001

Model 1: age, sex, principal components of genetic ancestry.

Model 2: Model 1 variables, genotype-predicted height.

Abbreviations: MESA = Multi-Ethnic Study of Atherosclerosis.

Supplementary Table 15: Association of height-GaP with mortality in UKBiobank and MESA using genotype-predicted height derived from cohort- and sex-specific regression models of measured height.

	Hazard ratio per 1-SD height-GaP deficit (95%CI) p-value	
	UKBiobank	MESA
All-cause death		
Model 1	1.11 (1.10, 1.13) p<0.001	1.10 (1.03, 1.17) p=0.005
Model 2	1.12 (1.10, 1.13) p<0.001	1.09 (1.01, 1.17) p=0.019
Atherosclerotic cardiovascular disease death		
Model 1	1.15 (1.13, 1.18) p<0.001	1.15 (0.99, 1.35) p=0.075
Model 2	1.15 (1.13, 1.18) p<0.001	1.19 (1.01, 1.40) p=0.038
Atherosclerotic coronary heart disease death		
Model 1	1.25 (1.22, 1.29) p<0.001	1.38 (1.12, 1.69) p=0.002
Model 2	1.25 (1.22, 1.29) p<0.001	1.35 (1.12, 1.64) p=0.002

Model 1: age, sex, principal components of genetic ancestry.

Model 2: Model 1 variables, genotype-predicted height.

Abbreviations: MESA = Multi-Ethnic Study of Atherosclerosis; SD = standard deviation.

R2.9. The methods section of the abstract (lines 79-80) describes variables that were used to adjust the regression models, but it does not mention what were the independent and dependent variables in these regression models.

The abstract text has been revised.

Abstract: "Regression models examined: i) adult height-GaP as the outcome with early-life growth conditions as predictors; and ii) mortality as the outcome with adult height-GaP as predictor."

R2.10. In Table 1, Age at height-GaP assessment: IQR is mentioned in the variable description but not 'median'

Apologies for this oversight. The row header text has been amended.

R2.11. Figure 2: Please explain the N's below the figures. I assume they are from the timepoints (0, 5, 10, 15 yrs) marked in the figure but this is not clear.

Your interpretation is correct.

Figure 2 text has been modified to improve clarity, adding a label to specify "Time Since Baseline (Years)."

R3.1. Lines 143-145: What's the difference between initial recruitment and further recruitment?

*Apologies for the lack of clarity. ALSPAC sought to enrol all pregnant women residing in the three Health Districts of Avon County if their estimated delivery date fell between April 1 1991 and December 31 1992, inclusive. Any resulting child from these pregnancies was considered eligible for the cohort study.³⁶ **Initial recruitment** occurred between 1990 and 1992 and was opportunistic, aiming to recruit women as early in pregnancy as possible.^{37,38} During this initial recruitment campaign, the ALSPAC target sample was a dynamic population for whom no convenient sampling frame was available to support systematic invitation of all eligible individuals. The eligible study sample was later defined by cross referencing maternity, birth and child health records with ALSPAC recruitment records. This necessarily retrospective process of defining the 'eligible sample' identified additional pregnancies where the offspring were eligible for recruitment, but for whom no replies to recruitment invitations in the 1990–92 initial recruitment had been received. **Further recruitment** of these eligible children not previously enrolled were performed at visit years 7 and 8 of ALSPAC.^{37,38}*

The Methods text has been revised to improve clarity.

Methods: "Initial recruitment of 14,541 pregnancies resulted in 13,988 children alive after 1 year. Following identification of additional eligible children born between April 1 1991 and December 31 1992 in Avon, England, further recruitment attempts at follow-up years 7 and 8 resulting in a total of 15,447 pregnancies included in the study, with 14,901 children alive after 1 year."

R3.2. Lines 159-162: How many were initially recruited?

N=1,037. The Methods text has been revised to improve clarity.

Methods: "The Dunedin Multidisciplinary Health and Development Study (DMHDS) is a population-based birth cohort of 1,037 participants born between April 1972 and March 1973 at Queen Mary Maternity Hospital in Dunedin, New Zealand"

R3.3. Lines 193-194: If I understand correctly, a lower height-GaP means that the measured adult height is close to the genotype-predicted height? Hence, a smaller deficit?

Height-GaP = attained adult height in cm – genotype-predicted height in cm

Thus, a lower height-GaP value signifies a larger deficit between attained adult height and genotype-predicted height. Why did we define height-GaP this way? We piloted various approaches to express the height-GaP construct (e.g., attained height – genotype-predicted height; genotype-predicted height – attained height; expressing all height-GaP values relative to the maximum observed height-GaP in the sample; expressing height-GaP in z-scores; etc...). While every approach yielded equivalent inferences with respect to early-life growth conditions and later-life mortality, we noted that each approach had some strengths and some limitations with respect to ease of communication. The current construct (height-GaP = attained adult height - genotype-predicted height) seemed to strike the optimum balance with respect to audience understanding of height-GaP calculation, meaning, and interpretation of association estimates.

R3.4. Lines 204-206: Is this a measure for socio-economic status? What do the authors mean by multiple deprivation during pregnancy?

The multiple deprivation index (MDI) is a composite measure used in some jurisdictions (e.g., U.K., New Zealand) to quantify multiple aspects of social disadvantage based on small geographic areas (e.g., postal code, census tract, etc.). Multiple area-level statistics including household income, employment, education, disability, crime and environment are combined into a standardized index where higher scores indicate greater deprivation. While the MDI includes some domains related to socio-economic status (e.g., indices of income, wealth), it is generally considered a broader index that captures additional aspects of deprivation (e.g., crime, environment).

“Multiple deprivation index during pregnancy” refers to an MDI computed using the residential address (and corresponding area-level statistics) of the mother during the pregnancy. The Methods text has been revised to improve clarity.

Methods: “Residential address information during pregnancy, at birth, and at post-natal ages 1-12 years were used to compute neighbourhood-level indices of multiple deprivation during in ALSPAC, with higher values corresponding to greater deprivation.”

RESPONSE REFERENCES

1. Jousilahti P, Tuomilehto J, Vartiainen E, Eriksson J, Puska P. Relation of Adult Height to Cause-specific and Total Mortality: A Prospective Follow-up Study of 31, 199 Middle-aged Men and Women in Finland. *American Journal of Epidemiology*. 2000;151(11):1112-1120. doi:10.1093/oxfordjournals.aje.a010155
2. Li Q, Liu Y, Sun X, et al. Dose–response association between adult height and all-cause mortality: a systematic review and meta-analysis of cohort studies. *European Journal of Public Health*. 2021;31(3):652-658. doi:10.1093/eurpub/ckaa213
3. Wadsworth MEJ, Hardy RJ, Paul AA, Marshall SF, Cole TJ. Leg and trunk length at 43 years in relation to childhood health, diet and family circumstances; evidence from the 1946 national birth cohort. *Int J Epidemiol*. 2002;31(2):383-390.
4. Li L, Dangour AD, Power C. Early life influences on adult leg and trunk length in the 1958 British birth cohort. *Am J Hum Biol*. 2007;19(6):836-843. doi:10.1002/ajhb.20649
5. Denholm R, Power C, Li L. Adverse childhood experiences and child-to-adult height trajectories in the 1958 British birth cohort. *Int J Epidemiol*. 2013;42(5):1399-1409. doi:10.1093/ije/dyt169
6. Gigante DP, Nazmi A, Lima RC, Barros FC, Victora CG. Epidemiology of early and late growth in height, leg and trunk length: findings from a birth cohort of Brazilian males. *Eur J Clin Nutr*. 2009;63(3):375-381. doi:10.1038/sj.ejcn.1602949
7. Yengo L, Vedantam S, Marouli E, et al. A saturated map of common genetic variants associated with human height. *Nature*. 2022;610(7933):704-712. doi:10.1038/s41586-022-05275-y
8. Chan Y, Salem RM, Hsu YHH, et al. Genome-wide Analysis of Body Proportion Classifies Height-Associated Variants by Mechanism of Action and Implicates Genes Important for Skeletal Development. *The American Journal of Human Genetics*. 2015;96(5):695-708. doi:10.1016/j.ajhg.2015.02.018
9. Kun E, Javan EM, Smith O, et al. The genetic architecture and evolution of the human skeletal form. *Science*. 2023;381(6655):eadf8009. doi:10.1126/science.adf8009
10. Fiorito G, Polidoro S, Dugué PA, et al. Social adversity and epigenetic aging: a multi-cohort study on socioeconomic differences in peripheral blood DNA methylation. *Sci Rep*. 2017;7(1):16266. doi:10.1038/s41598-017-16391-5
11. Lussier AA, Zhu Y, Smith BJ, et al. Association between the timing of childhood adversity and epigenetic patterns across childhood and adolescence: findings from the Avon Longitudinal Study of Parents and Children (ALSPAC) prospective cohort. *The*

Lancet Child & Adolescent Health. 2023;7(8):532-543. doi:10.1016/S2352-4642(23)00127-X

12. Pearce MS, Mann KD, Martin-Ruiz C, et al. Childhood growth, IQ and education as predictors of white blood cell telomere length at age 49-51 years: the Newcastle Thousand Families Study. *PLoS One*. 2012;7(7):e40116. doi:10.1371/journal.pone.0040116

13. Masterson EE, Hayes MG, Kuzawa CW, Lee NR, Eisenberg DTA. Early life growth and adult telomere length in a Filipino cohort study. *Am J Hum Biol*. 2019;31(6):e23299. doi:10.1002/ajhb.23299

14. Kuzawa CW, Ryan CP, Adair LS, et al. Birth weight and maternal energy status during pregnancy as predictors of epigenetic age acceleration in young adults from metropolitan Cebu, Philippines. *Epigenetics*. 2022;17(11):1535-1545. doi:10.1080/15592294.2022.2070105

15. Maddock J, Castillo-Fernandez J, Wong A, et al. Childhood growth and development and DNA methylation age in mid-life. *Clinical Epigenetics*. 2021;13(1):155. doi:10.1186/s13148-021-01138-x

16. Kuula J, Czamara D, Hauta-alus H, et al. Epigenetic signature of very low birth weight in young adult life. *Pediatr Res*. 2025;97(1):229-238. doi:10.1038/s41390-024-03354-6

17. Kuh D, Susser E, Blodgett JM, Ben-Shlomo Y. *A Life Course Approach to the Epidemiology of Chronic Diseases and Ageing*. 3rd ed. Oxford University Press; 2025.

18. Barker DJ, Osmond C. Infant mortality, childhood nutrition, and ischaemic heart disease in England and Wales. *Lancet*. 1986;1(8489):1077-1081. doi:10.1016/s0140-6736(86)91340-1

19. Barker DJ, Winter PD, Osmond C, Margetts B, Simmonds SJ. Weight in infancy and death from ischaemic heart disease. *Lancet*. 1989;2(8663):577-580. doi:10.1016/s0140-6736(89)90710-1

20. Eriksson JG, Forsén T, Tuomilehto J, Osmond C, Barker DJ. Early growth, adult income, and risk of stroke. *Stroke*. 2000;31(4):869-874. doi:10.1161/01.str.31.4.869

21. Fall CH, Barker DJ, Osmond C, Winter PD, Clark PM, Hales CN. Relation of infant feeding to adult serum cholesterol concentration and death from ischaemic heart disease. *BMJ*. 1992;304(6830):801-805. doi:10.1136/bmj.304.6830.801

22. Martyn CN, Barker DJ, Osmond C. Mothers' pelvic size, fetal growth, and death from stroke and coronary heart disease in men in the UK. *Lancet*. 1996;348(9037):1264-1268. doi:10.1016/s0140-6736(96)04257-2

23. Risnes KR, Vatten LJ, Baker JL, et al. Birthweight and mortality in adulthood: a systematic review and meta-analysis. *Int J Epidemiol*. 2011;40(3):647-661. doi:10.1093/ije/dyq267
24. Syddall HE, Sayer AA, Simmonds SJ, et al. Birth Weight, Infant Weight Gain, and Cause-specific Mortality: The Hertfordshire Cohort Study. *American Journal of Epidemiology*. 2005;161(11):1074-1080. doi:10.1093/aje/kwi137
25. Kajantie E, Osmond C, Barker DJP, Forsén T, Phillips DIW, Eriksson JG. Size at birth as a predictor of mortality in adulthood: a follow-up of 350 000 person-years. *Int J Epidemiol*. 2005;34(3):655-663. doi:10.1093/ije/dyi048
26. Osmond C, Barker DJ, Winter PD, Fall CH, Simmonds SJ. Early growth and death from cardiovascular disease in women. *BMJ*. 1993;307(6918):1519-1524. doi:10.1136/bmj.307.6918.1519
27. Hu L, Wu S, Zhang Y, et al. Associations of maternal and personal smoking with all-cause and cause-specific mortality risk and life expectancy: a prospective cohort study. *Public Health*. 2024;229:144-150. doi:10.1016/j.puhe.2024.02.003
28. Diver WR, Jacobs EJ, Gapstur SM. Secondhand Smoke Exposure in Childhood and Adulthood in Relation to Adult Mortality Among Never Smokers. *Am J Prev Med*. 2018;55(3):345-352. doi:10.1016/j.amepre.2018.05.005
29. Teramoto M, Iso H, Muraki I, Shirai K, Tamakoshi A. Secondhand Smoke Exposure in Childhood and Mortality from Coronary Heart Disease in Adulthood: the Japan Collaborative Cohort Study for Evaluation of Cancer Risk. *J Atheroscler Thromb*. 2023;30(8):863-870. doi:10.5551/jat.63857
30. Wang X, Yan M, Zhang Y, et al. Breastfeeding in infancy and mortality in middle and late adulthood: A prospective cohort study and meta-analysis. *J Intern Med*. 2023;293(5):624-635. doi:10.1111/joim.13619
31. Crump C, Sundquist K, Sundquist J, Winkleby MA. Gestational age at birth and mortality in young adulthood. *JAMA*. 2011;306(11):1233-1240. doi:10.1001/jama.2011.1331
32. Risnes K, Bilsteen JF, Brown P, et al. Mortality Among Young Adults Born Preterm and Early Term in 4 Nordic Nations. *JAMA Netw Open*. 2021;4(1):e2032779. doi:10.1001/jamanetworkopen.2020.32779
33. Smith GD, Hart C, Blane D, Hole D. Adverse socioeconomic conditions in childhood and cause specific adult mortality: prospective observational study. *BMJ*. 1998;316(7145):1631-1635. doi:10.1136/bmj.316.7145.1631

34. Frankel S, Smith GD, Gunnell D. Childhood socioeconomic position and adult cardiovascular mortality: the Boyd Orr Cohort. *Am J Epidemiol.* 1999;150(10):1081-1084. doi:10.1093/oxfordjournals.aje.a009932
35. Power C, Hyppönen E, Davey Smith G. Socioeconomic Position in Childhood and Early Adult Life and Risk of Mortality: A Prospective Study of the Mothers of the 1958 British Birth Cohort. *Am J Public Health.* 2005;95(8):1396-1402. doi:10.2105/AJPH.2004.047340
36. Fraser A, Macdonald-Wallis C, Tilling K, et al. Cohort Profile: The Avon Longitudinal Study of Parents and Children: ALSPAC mothers cohort. *International Journal of Epidemiology.* 2013;42(1):97-110. doi:10.1093/ije/dys066
37. Boyd A, Golding J, Macleod J, et al. Cohort Profile: The 'Children of the 90s'—the index offspring of the Avon Longitudinal Study of Parents and Children. *International Journal of Epidemiology.* 2013;42(1):111-127. doi:10.1093/ije/dys064
38. Northstone K, Lewcock M, Groom A, et al. The Avon Longitudinal Study of Parents and Children (ALSPAC): an update on the enrolled sample of index children in 2019. *Wellcome Open Res.* 2019;4:51. doi:10.12688/wellcomeopenres.15132.1